# Folic Acid Supplementation Promotes Hypomethylation in Both the Inflamed Colonic Mucosa and Colitis-Associated Dysplasia

**DOI:** 10.3390/cancers15112949

**Published:** 2023-05-27

**Authors:** Wen-Chi L. Chang, Jayashri Ghosh, Harry S. Cooper, Lisa Vanderveer, Bryant Schultz, Yan Zhou, Kristen N. Harvey, Esther Kaunga, Karthik Devarajan, Yuesheng Li, Jaroslav Jelinek, Mariana F. Fragoso, Carmen Sapienza, Margie L. Clapper

**Affiliations:** 1Cancer Prevention and Control Program, Fox Chase Cancer Center, 333 Cottman Avenue, Philadelphia, PA 19111, USAlisa.vanderveer@fccc.edu (L.V.); ek2941@cumc.columbia.edu (E.K.);; 2Fels Cancer Institute for Personalized Medicine, Lewis Katz School of Medicine, Temple University, 3500 N. Broad Street, Philadelphia, PA 19140, USA; jayashri.ghosh@temple.edu (J.G.); bryant.schultz@fccc.edu (B.S.);; 3Department of Pathology, Fox Chase Cancer Center, 333 Cottman Avenue, Philadelphia, PA 19111, USA; 4Biostatistics and Bioinformatics Facility, Fox Chase Cancer Center, 333 Cottman Avenue, Philadelphia, PA 19111, USA; 5DNA Sequencing and Genomic Core Facility, National Heart, Lung, and Blood Institute, NIH, 10 Center Drive, Bethesda, MD 20892, USA

**Keywords:** folic acid, colitis, methylation, colon, tumorigenesis

## Abstract

**Simple Summary:**

Ulcerative colitis, a form of inflammatory bowel disease, is a major risk factor for developing colorectal cancer. Patients with ulcerative colitis often present with low serum folate levels due to the adverse effect of maintenance therapies and/or intestinal malabsorption and require folic acid supplementation. While folic acid supplementation confers protection against sporadic colorectal cancer when given prior to colon tumor formation, it can promote cancer formation when administered to mice with early colonic lesions. Few studies have focused on the impact of folic acid on colorectal cancer risk in patients with ulcerative colitis. In this study, folic acid supplementation created a hypomethylated field within the inflamed non-neoplastic colonic mucosa of mice with colitis and promoted the formation of colitis-associated tumors. These data suggest that caution should be taken when recommending folic acid supplements to patients with ulcerative colitis who are at high risk of colorectal cancer.

**Abstract:**

Purpose: The purpose of this study was to assess the effect of folic acid (FA) supplementation on colitis-associated colorectal cancer (CRC) using the azoxymethane/dextran sulfate sodium (AOM/DSS) model. Methods: Mice were fed a chow containing 2 mg/kg FA at baseline and randomized after the first DSS treatment to receive 0, 2, or 8 mg/kg FA chow for 16 weeks. Colon tissue was collected for histopathological evaluation, genome-wide methylation analyses (Digital Restriction Enzyme Assay of Methylation), and gene expression profiling (RNA-Seq). Results: A dose-dependent increase in the multiplicity of colonic dysplasias was observed, with the multiplicity of total and polypoid dysplasias higher (64% and 225%, respectively) in the 8 mg FA vs. the 0 mg FA group (*p* < 0.001). Polypoid dysplasias were hypomethylated, as compared to the non-neoplastic colonic mucosa (*p* < 0.05), irrespective of FA treatment. The colonic mucosa of the 8 mg FA group was markedly hypomethylated as compared to the 0 mg FA group. Differential methylation of genes involved in Wnt/β-catenin and MAPK signaling resulted in corresponding alterations in gene expression within the colonic mucosa. Conclusions: High-dose FA created an altered epigenetic field effect within the non-neoplastic colonic mucosa. The observed decrease in site-specific DNA methylation altered oncogenic pathways and promoted colitis-associated CRC.

## 1. Introduction

Patients with ulcerative colitis face a cumulative risk of developing colorectal cancer (CRC) that is significantly higher than that of the general population, 5–10% after 20 years and 15–20% after 30 years of disease [1]. The association of disease duration, severity, and extent [2] with colon cancer risk provides strong support for the critical role of inflammation in colitis-associated CRC [3]. Patients with colitis often present with serum folate levels that are 15–20% lower than those of healthy individuals [4,5]. Explanations for this deficiency include medications that inhibit FA transport or malabsorption from the loss of intestinal surface area. Sulfasalazine, a common therapy for the maintenance of patients with ulcerative colitis, inhibits the absorption of folate in the intestinal mucosa [6]. While FA supplements are often recommended for colitis patients, the risk vs. benefit of giving FA remains unclear based on the potential association of FA supplement use with the risk of sporadic CRC.

Results from several epidemiologic studies have indicated that the risk of developing CRC is decreased significantly among subjects taking FA supplements [7,8]. In contrast, an emerging body of evidence suggests that FA supplementation does not confer protection against colorectal tumorigenesis, especially in individuals with a strong underlying predisposition for CRC (i.e., history of adenomas) [9,10,11]. In fact, administration of FA (1 mg/day for 6–8 years) to subjects with a history of adenomas in a double-blinded, placebo-controlled, randomized clinical trial increased the risk of developing both advanced tumors (RR 1.67, *p* = 0.05) and multiple (≥3) colorectal adenomas (RR 2.32, *p* = 0.02) [9]. Results from a prospective cohort study [12] revealed a direct association between plasma folate levels and risk of CRC in subjects who were followed for longer than the median 4.2 years (*p* trend = 0.007). In mice, the ability of FA to promote CRC development is dependent upon the timing and dose administered [13,14,15,16]. Diets supplemented with 8 mg FA/kg decreased tumor burden when given to Apc^+/−^/Msh2^−/−^ mice before the establishment of aberrant crypt foci, a surrogate biomarker of CRC. However, the same diet increased intestinal tumor burden when administered to mice with existing aberrant crypt foci [14].

Few studies have investigated the impact of FA supplementation on colitis-associated CRC. Results from two case-control studies [17,18] and one cohort study [19] suggested that colitis patients who took FA supplements had a lower adjusted relative risk of developing colitis-associated neoplasia, as compared to colitis patients not taking supplements. However, this finding did not achieve statistical significance. Compromising characteristics of these retrospective analyses include the small sample size, lack of information on the dose of FA employed, length of exposure, and corresponding plasma folate levels. Interestingly, red blood cell folate levels are reported to be 19% higher in children with untreated inflammatory bowel disease, as compared to healthy controls [20]. Based on these inconsistent observations, further investigation of the relationship between inflammation, folate status, and CRC is warranted.

The overall goal of the present study was to assess the effect of varying concentrations of FA on the development of colitis-associated neoplasia in a model that recapitulated the clinical setting in which patients take a FA supplement following a diagnosis of colitis. In addition to histopathological enumeration of dysplasias, dose-associated FA alterations in DNA methylation were assessed and compared to gene expression changes in both the non-dysplastic inflamed colonic mucosa and colitis-associated neoplastic lesions. The resulting data demonstrate that FA supplementation promotes colitis-associated CRC by creating a hypomethylated field within the non-neoplastic colonic mucosa. Pathway analyses of genes differentially methylated among groups administered varying levels of FA revealed many genes associated with colorectal carcinogenesis. When combined, these novel data suggest that FA induces a stress response in the presence of chronic inflammation, leading to decreased site-specific DNA methylation of genes that promote the development of colitis-associated tumors.

## 2. Materials and Methods

### 2.1. Diet and Animal Treatment

All animal studies were approved by the Institutional Animal Care and Use Committee at Fox Chase Cancer Center. Diets were purchased from Bio-Serv (Flemington, NJ, USA). The control AIN-76A diet contained 2 mg FA/kg chow (2 mg FA) which is equivalent to a daily intake of 400 μg/day in humans. The deficient diet was a modified AIN-76A diet containing no FA (0 mg FA), and the supplemented diet was a modified AIN-76A diet containing 8 mg FA/kg chow (8 mg FA). Mouse chow with 8 mg/kg FA is equivalent to a daily intake of 1.6 mg FA by humans and comparable to the level consumed by clinical trial participants who take a FA supplement (1 mg/day) and continue to ingest FA-fortified foods.

The AOM/DSS model of colitis-associated colorectal carcinogenesis was employed in this study according to Figure 1. At 6 weeks of age, female Swiss Webster mice (Taconic Farms, Germantown, NY, USA) (17–19/group) were fed the AIN-76A control diet (2 mg FA). Mice were injected with AOM (7.4 mg/kg i.p.; Sigma-Aldrich, St Louis, MO, USA) at 7 weeks of age. Mice without DSS served as non-colitic controls (*n* = 8/group). At 9 weeks of age, colitis was induced by administering DSS (MW 40,000; Alfa Aesar, Tewksbury, MA) for 4 cycles, with each cycle consisting of 7 days of 2% DSS in the drinking solution followed by 14 days of untreated water. At 11 weeks of age and immediately after the first DSS treatment (acute colitis phase), mice were randomized to receive FA at 0 (deficient), 2, or 8 (high dose) mg/kg of chow for 16 weeks. Body weights were recorded weekly. At 27 weeks of age, the mice were euthanized and their colons collected, fixed in 10% formalin, cross-sectioned at 2 mm intervals, and processed in their entirety for histopathological evaluation. 

### 2.2. Histopathology

Sections stained with hematoxylin and eosin (H&E) were evaluated in a blinded manner as described previously [21]. Tissue specimens were classified as negative or positive for dysplasia or carcinoma based on the standardized morphology and nomenclature for the human pathology of colitis-associated CRC [22]. A diagnosis of carcinoma was assigned when neoplastic glands had invaded beyond the muscularis mucosa and into the submucosa. Carcinomas and dysplasias were classified as polypoid (elevated growth pattern) or flat (no elevated growth component and less than twice the height of the adjacent non-dysplastic mucosa) (Figure 2).

### 2.3. Isolation of DNA and RNA from Laser Microdissected (LMD) Colonic Epithelial Cells 

Cells were microdissected from the paraffin-embedded non-dysplastic colon and dysplasia on PEN membrane slides (Life Technologies, Waltham, MA, USA) using a Leica LMD6500 system (Leica Microsystems, Deerfield, IL, USA). DNA and RNA were isolated from LMD samples using RecoverAll™ Total Nucleic Acid Isolation for FFPE (Life Technologies), according to the manufacturer’s protocol. RNA concentration was determined and quality assessed on an Agilent 2100 Bioanalyzer using RNA 6000 Pico chips (Agilent Technologies, Santa Clara, CA, USA).

### 2.4. Genome-Wide CpG island Methylation Analysis

Genome-wide DNA methylation analyses were performed on LMD colonic epithelial cells or mucosa using the Digital Restriction Enzyme Analysis of Methylation (DREAM) [23] assay. Briefly, the DREAM assay utilizes a combination of methylation-sensitive (SmaI) and methylation-insensitive (XmaI) restriction endonucleases that recognize the CCCGGG sequence, but cleave at different locations within this target sequence. Methylated and unmethylated sites are distinguished by virtue of the distinct 5′-/3′-end sequence. Quantitation is achieved by deep sequencing on the Illumina HiSeq^®^ instrument and mapping the reads back to the human genome using a custom script [23]. DNA (approximately 500 ng) was isolated from LMD material, as described above, and analyzed. 

### 2.5. RNA-Seq Analysis 

RNA-Seq analyses were performed on LMD colonic epithelial cells. RNA-Seq libraries were generated using a SMARTer® Stranded RNA-Seq Kit V2 Pico Input Mammalian (TakaRa Bio USA, Inc. Mountain View, CA, USA) with 40 ng total RNA. Libraries were quantified using a Qubit^TM^ dsDNA HS Kit (Thermo Fisher Scientific, Waltham, MA, USA) and size distribution validated on an Agilent 2100 Bioanalyzer using the Agilent High Sensitivity DNA Kit (Agilent Technologies). Validated libraries were sequenced on an Illumina HiSeq 2500 platform and analyzed. The resulting sequence reads were analyzed to identify differentially expressed genes, using two methods, and genes found in common were overlayed with methylation changes as a secondary step. The first method involved aligning RNA-Seq data using Bowtie [24] to a bowtie-indexed mouse mm10 genome. The number of raw counts in each known gene from the RefSeq database was enumerated using htseq-count from the HTSeq package [25]. Differential expression between samples and across different conditions were assessed for statistical significance using the R/Bioconductor package DESeq2 [26]. Genes with a false discovery rate (FDR) ≤0.05, calculated using the Benjamini–Hochberg FDR method [27], and a fold-change ≥2 were considered significant. In the second method, Tophat2 [28] was used to align reads to the mouse mm10 genome. The Cufflinks [29] algorithm was implemented to assemble transcripts and estimate their abundance. Cuffdiff [30] was employed to statistically assess expression changes in quantified genes under different conditions. Genes with a false discovery rate of ≤5% and a fold change ≥1.5 were considered differentially expressed.

### 2.6. Immunohistochemistry (IHC) of β-Catenin

Expression of β-catenin was detected in formalin-fixed paraffin-embedded sections of colon tissue. Rabbit anti-β-catenin antibody was purchased from Sigma-Aldrich (St. Louis, MO, USA) and diluted to 1:2000 prior to use. Tissue sections were stained using a Roche Ventana Discovery XT automated staining instrument (Ventana Medical Systems, Tucson, AZ, USA) and Ventana reagents according to the manufacturer’s instructions. The primary antibody was replaced with normal rabbit IgG to serve as a negative control.

### 2.7. Real-Time Quantitative PCR (RT-qPCR)

Expression of select genes associated with colitis was evaluated in an independent set of LMD colonic mucosa samples from mice treated with different levels of FA using TaqMan assays (Thermo Fisher Scientific) and TaqMan^TM^ Gene Expression Master Mix (Thermo Fisher Scientific). The IDs for these TaqMan assays were the following: *IL-1β* Mm00434228_m1; *IL-6* Mm00446190_m1; *IL-10* Mm01288386_m1; *IL-17α* Mm00439618_m1; *IL-23α* Mm00518984_m1; *Ikbkb* Mm01222247_m1; *Mmp9* Mm00442991_m1; *Ptgs-2* Mm00478374_m1; and *Actb (β-actin*) Mm00607939_s1. *β-actin* was used as an endogenous control. Amplification products were monitored using a ABI7900 Sequence Detection System (Thermo Fisher Scientific). The resulting data were analyzed and expressed as the target gene expression relative to the endogenous control, using the comparative Ct method and the 2^−ΔΔCt^ formula. Results were expressed as the fold change in relative levels of each gene transcript for mice receiving different levels of FA.

### 2.8. Biostatistical Analyses

A generalized linear model approach (Poisson regression with log link) was used to determine the association between the FA dose and multiplicity of colonic dysplasias. Adjustment for multiple testing was performed using the Benjamini–Hochberg FDR method [27] and computations were performed using the R language [31]. Statistical comparisons of site-specific DNA methylation levels between FA treatment groups (with and without DSS) were performed using *t*-tests. Sites with *p* values less than 0.05 and a more than 5% methylation difference were considered significantly different between groups.

## 3. Results

### 3.1. Treatment Tolerance 

Body weights were monitored weekly throughout the experiment. No significant differences in mean body weight were observed among the treatment groups (*p* > 0.05) (Figure 3). The highest study completion rate was observed in the 0 mg FA group (89.5%, 17/19), followed by the 8 mg FA group (77.8%, 14/18), with the lowest rate in the 2 mg FA group (64.7%, 11/17). All mice receiving only AOM (no DSS) completed the experiment. 

### 3.2. Effect of FA on the Development of Colitis-Associated Dysplasia and Cancer

A dose-dependent increase in the multiplicity of total colitis-associated dysplasia was as observed with increasing levels of dietary FA (*p* < 0.0001) (Figure 4). The total number of dysplasias per mouse was 24% and 65.6% higher in the 2 and 8 mg FA groups, respectively, as compared to the 0 mg FA group. The multiplicity of polypoid dysplasias was 53% and 225% higher in the 2 mg FA and 8 mg FA groups, respectively, as compared to that of the 0 mg FA group (*p* < 0.001). However, only mice fed the 8 mg FA diet had an increased number of flat dysplasias (47.7% higher; *p* = 0.035). Only one carcinoma was observed in the entire study, and this mouse was treated with 8 mg FA. No dysplasias/cancers were found in AOM-treated (no DSS) mice. 

### 3.3. Effect of FA on Genome-Wide DNA Methylation at CCCGGG Sites 

Genome-wide DNA methylation analyses were performed using two types of LMD samples: (1) non-dysplastic epithelial cells from animals treated with AOM/DSS or only AOM and (2) the epithelium plus the lamina propria from AOM/DSS-treated mice. The latter was used due to a greater yield of DNA. Both types of samples gave equivalent results (see below). 

*DNA methylation in **non-dysplastic** colonic epithelial cells*. A dose-dependent, genome-wide effect of FA on DNA methylation was observed in LMD non-dysplastic colonic epithelial cells (Figure 5). With respect to repeat sequences of DNA, the 8 mg FA and 2 mg FA groups had more CpG sites with significant methylation changes (22.6% and 14.1%, respectively) than the 0 mg FA group; over 99% of these changes were hypomethylation (Figure 5A,B). In addition, the hypomethylating effect of FA on non-dysplastic colonic epithelial cells was dose-dependent. In unique sequences, the number of significant methylation changes in CpGs was only slightly higher in the 8 mg FA and 2 mg FA groups (2.7% and 1.5%, respectively) than in the 0 mg FA group (Figure 5C,D). Although fewer CpGs were modified by FA in unique vs. repeat sequences, hypomethylation remained the main outcome. Following supplementation, hypomethylation of CpG sites in unique sequences was elevated 80.7% and 61.5% in the 8 mg FA and 2 mg FA groups, respectively, over that of the 0 mg FA group (Figure 5C,D). This hypomethylating effect of FA on both repeat and unique sequences in LMD non-dysplastic epithelial cells was confirmed using the LMD epithelium plus the lamina propria (Figure 6; 8 mg FA vs. 0 mg FA results shown). Of note, the hypomethylating effect of FA was limited to mice with AOM/DSS-induced colitis. No substantial hypomethylating effect of FA was observed in mice treated with only AOM (no DSS) (Figure 7), indicating that the DNA methylation differences in the non-dysplastic colonic epithelium occur only in the context of inflammation. 

*DNA methylation in **dysplastic** tissues.* Although both polypoid and flat dysplasia were observed in all treatment groups (Figure 4), further analyses were restricted to polypoid dysplasia from the 8 mg FA and 0 mg FA groups, as these lesions exhibited the largest and most statistically significant inter-group difference in the mean multiplicity of dysplasia. For repeat sequences, there were 16.2% more CpGs with significant methylation changes in polypoid lesions treated with 8 mg FA (8P), as compared to the non-dysplastic epithelium exposed to 8 mg FA (8ND) (Figure 8A). Likewise, the methylation changes in polypoid dysplasia from the 0 mg FA (0P) group were 9.7% higher than in the 0 mg FA non-dysplastic colonic mucosa (0ND) (Figure 8B). More than 93% of the identified CpGs with significant methylation changes in repeat sequences (96.3% for 8 mg FA and 93.9% for 0 mg FA) were hypomethylated (Figure 8A,B). Unlike the repeat sequences, a lower percentage of CpGs with significant methylation changes, specifically hypomethylation, was observed in the unique sequences. In the unique sequences, there were 6.5% more CpGs with significant methylation changes in 8P lesions, as compared to the non-neoplastic mucosa (8ND) (Figure 8C). The methylation changes in 0P lesions were 7.3% higher than in the corresponding non-neoplastic mucosa (0ND) (Figure 8D). Less than 80% of the identified CpGs with significant methylation changes in unique sequences were hypomethylated (Figure 8C,D). The CpG methylation changes in lesions in the 8P and 0P groups were also compared (Figure 8E,F). The percentage of CpGs with significant methylation changes in 8P lesions was slightly higher than that of 0P lesions for both repeat (Figure 8E) and unique (Figure 8F) sequences. These data indicate that (1) 8P had the most changes in CpG methylation (hypomethylation specifically) in repeat sequences as compared to those in the 8ND and (2) fewer significant differences in DNA methylation were found when comparing 8P vs. 0P. These observations suggest that many of the epigenetic changes that accompany the development of polypoid dysplasia occur irrespective of the colonic mucosal environment in which they arise. 

To gain insight into the FA-induced methylation changes that accompany cellular transformation, the DNA methylation profile of 0P lesions was compared to that of 8ND epithelium (Figure 9A,B). Relatively few significant differences in methylation were detected in both repeat (3.9%) and unique (3.8%) sequences in the 0P vs. 8ND comparison, and no strong bias towards hypomethylation (64.8% in repeat sequences and 48.6% in unique sequences) was observed (Figure 9A,B). This suggests that supplementation with 8 mg FA modifies the methylation status of the background colonic mucosa to become more “tumor-like”. In contrast, the highest percentage of CpGs with significant methylation changes was observed when comparing 8P lesions vs. 0ND epithelium; a 40.3% increase in repeat sequences (Figure 9C) and a 12% increase in unique sequences (Figure 9D) was observed. In addition, 8P lesions were strongly hypomethylated, 81.2% in repeat and 78.1% in unique sequences, as compared to the 0ND epithelium (Figure 9C,D). These data, when compared with the DNA methylation profile of 0P lesions vs. 0ND epithelium (Figure 8B,D), indicate that the 0ND epithelium exhibits the fewest changes in DNA methylation of all the groups evaluated.

### 3.4. Expression of Genes whose Methylation Status was Modified by FA Treatment

Based on the dose-dependent effect of FA on the formation of colitis-associated dysplasia and genome-wide DNA methylation, gene expression analyses were performed using tissues from only the 0 mg FA and 8 mg FA groups. Both non-dysplastic colonic epithelial cells (8ND and 0ND) and neoplastic cells from polypoid dysplasia (8P and 0P) were LMD and processed for RNA-Seq analyses. To enrich for gene expression differences associated with FA supplementation, analyses were restricted to expression changes in genes whose DNA methylation levels differed in 8ND vs. 0ND epithelial cells. Of the 411 genes whose methylation levels differed significantly between these two groups (Appendix A), only 15 genes were identified as differentially expressed by RNA-Seq (Table 1). Similar changes in the transcript levels of these 15 genes were not observed when comparing 8P vs. 0P lesions (Table 1). The 15 genes differentially expressed following FA supplementation were categorized into three major signaling pathways: Wnt/β-catenin (Tbxt, Zfp703, Enc1, Tns4, and Neu1), MAPK (Dusp4 and Epha2), and cell proliferation and differentiation (Spink4, Thbs1, Lzts1, and Fut9). Among these genes, Dusp4, Epha2, Atp11a, and Tns4 are reported to be overexpressed in human CRC, while the expression of Thbs1 is increased in patients with inflammatory bowel disease (see Table 1). Interestingly, 12 of the 15 genes (except Enc1, Atp11a, and Lzts1) differentially expressed in the 8ND vs. 0ND groups (Table 1, Column B) also exhibited a similar trend of change in polypoid dysplasia vs. non-dysplastic mucosa (Table 1, Columns D–E). Considering their known oncogenic functions, early DNA alterations in these genes may create a pro-tumorigenic environment in the non-dysplastic mucosa and be essential for colitis-associated tumorigenesis in the AOM-DSS model. 

Based on the strong representation of genes involved in β-catenin and MAPK signaling in Table 1, RNA-Seq data were further analyzed enriching for gene expression changes associated with these two pathways. The resulting heatmaps demonstrate the upregulation of a broader network of genes involved in β-catenin and MAPK signaling in 8P vs. 0P lesions (Figure 10). Because FA had the greatest effect on β-catenin signaling, these data were confirmed by comparing the cellular localization of β-catenin in non-dysplastic vs. dysplastic colon tissue, in the presence and absence of FA by immunohistochemical staining (Figure 11). As expected, β-catenin was located primarily at the cellular membrane of non-dysplastic colonic cells. In contrast, tumors exhibited both nuclear localization and overexpression. Interestingly, β-catenin was expressed in the majority of tumor cells of 8P lesions (average 197/281 = 70.1%), but to a lesser extent in those of 0P lesions (average 117/275 = 42.5%). These observations confirm the elevated expression of genes associated with the β-catenin network in 8P lesions, as compared to 0P lesions. 

### 3.5. Effect of FA on Inflammatory Markers

The impact of FA supplementation on the expression of genes associated with colitis was assessed by RT-qPCR. Non-neoplastic regions of the colonic mucosa and submucosa of AOM/DSS-treated mice fed diets containing 0, 2, or 8 mg FA (8 per group), confirmed histopathologically to be free of dysplasia, were microdissected for analysis. Genes evaluated included *IL-1β*, *IL-6*, *IL-10*, *IL-17α*, *IL-23α*, *Ikbkb*, *Mmp9*, *Ptgs2 (Cox-2),* and the reference gene *β-actin*. Of all markers evaluated, *IL-6* was most affected by FA concentration (Figure 12). Expression of *IL-6* in animals administered 2 and 8 mg FA was ≥2-fold higher than that of those in the FA-deficient (0 mg FA) group (2 mg vs. 0 mg: *p* = 0.05; 8 mg vs. 0 mg: *p* = 0.046). Treatment with 8 mg FA caused a 75% increase in *Ikbkb* expression, a surrogate of NFκB/Rela signaling, over that of animals maintained on a diet containing 2 mg FA (*p* = 0.006). No significant difference in *Ikbkb* RNA expression was observed among animals receiving 2 mg and 0 mg FA. Expression of *Ptgs2* was elevated 2-fold in mice fed 8 mg vs. 0 mg FA (*p* = 0.098). Expression of *IL-10*, *IL-17α*, or *IL-23α* mRNA in the non-neoplastic colonic epithelium was not impacted significantly by changes in the level of FA (*p* > 0.05). These data indicate the ability of FA supplementation to induce the expression of specific genes associated with inflammation in the non-neoplastic colonic mucosa.

## 4. Discussion 

The present study represents the first demonstration of a link between FA-induced colon DNA hypomethylation and the promotion of colitis-associated tumorigenesis. Notably, depletion or supplementation of dietary FA was initiated immediately after the induction of acute colitis to mimic clinical treatment. FA supplementation induced not only a dose-dependent increase in colitis-associated dysplasia, but also global DNA hypomethylation of both the non-dysplastic mucosa and dysplastic lesions. Importantly, FA-induced DNA hypomethylation was not observed in colonic epithelial cells of non-colitic controls. Based on the reported association between colonic DNA hypomethylation in the non-dysplastic mucosa and increased risk for CRC [53], these novel observations suggest that high-dose FA supplementation creates a global hypomethylated environment within the non-dysplastic colonic mucosa. This epigenetic change could lead to a further decrease in the site-specific DNA methylation of genes that promote the development of colitis-associated dysplasia. In the present study, supplementation with 8 mg FA during acute colitis led to an increased multiplicity of colonic dysplasias. This result is alarming based on the routine FA supplementation of colitis patients who present with low levels of plasma folate.

The impact of FA on colonic DNA methylation and its association with sporadic CRC remains controversial [54,55]. In general, FA supplementation can have either a protective or promotional effect on sporadic colon tumorigenesis [56,57]. Initiation of FA supplementation prior to the establishment of dysplasia inhibited the formation of sporadic CRC, with increased DNA methylation suggested as a possible mechanism. In contrast, FA supplementation failed to confer protection in the presence of a strong underlying predisposition for sporadic CRC. To our knowledge, no randomized clinical trials have examined the effect of FA supplementation on DNA methylation in the context of colitis-associated CRC, and few animal studies have evaluated the impact of FA on colitis-associated CRC. MacFarlane and colleagues [58] reported that administration of FA (0, 2, and 8 mg) to AOM/DSS-treated mice failed to alter the total number of colon tumors significantly but increased the incidence of adenocarcinomas and multiplicity of distal colon tumors. Unlike the present study, animals received FA supplementation prior to the induction of colitis and DNA methylation was not assessed. The ability of FA supplementation to promote global DNA hypomethylation has been observed in other organs (e.g., brain, liver, and kidney) following treatment with 8 mg FA [59]. These data, although limited, support the hypothesis that FA supplementation increases the risk of colitis-associated CRC by reducing DNA methylation and provide strong justification for further investigating the safety of using FA supplements in colitis patients. 

The nonspecific effect of FA supplementation on the epigenome may cause loss of DNA methylation at many sites and hypermethylation at relatively few sites (Figure 5 and Figure 9). Genes with CpG sites that are differentially methylated in the non-neoplastic epithelium, in response to varying levels of FA, include several genes known to promote CRC (Table 1). One explanation for the greater number of colonic dysplasias in animals fed 8 mg vs. 0 mg FA is that a much larger fraction of colonic epithelial cells undergoes CRC-associated gene hypomethylation in mice exposed to high-dose FA. Our proposed model (Figure 13) suggests that the number of epigenetic changes required for the non-dysplastic mucosa to become a polypoid dysplasia is relatively large, while the same transition in animals fed the 8 mg FA diet requires fewer epigenetic modifications. Consistent with this view, relatively few significant differences were identified when the global DNA methylation profile of 0P lesions vs. 8ND epithelium was compared (Figure 9A,B). In contrast, the greatest number of significant methylation differences was detected when comparing 8P lesions vs. 0ND epithelium (Figure 9C,D). In addition, tumors exhibited substantial hypomethylation as compared to the non-dysplastic mucosa under conditions of folate deficiency and high-dose supplementation, suggesting the hypomethylating effects were stronger in the dysplastic lesions. The only exception was the unique sequences in the 8 mg FA group, in which the dysplasia contained nearly equivalent numbers of hyper- and hypomethylated sites. One explanation for this discrepancy is that the 8ND epithelium is already hypomethylated (compared to the 0ND epithelium); hence, further hypomethylation of dysplasia may be difficult to discern. The hypomethylating effect of FA supplementation was greater in repeat sequences vs. unique sequences. Repeat sequences also showed a clear trend of hypomethylation in all dysplasias, irrespective of FA dose. It is noteworthy that hypomethylation in repetitive elements has been linked to increased mortality from cancer [60].

Effort was invested to uncover the FA-induced epigenetic changes responsible for the relative increase in polypoid dysplasia. If the genes identified as being hypomethylated are involved in the formation of polypoid tumors, one would expect them to also be differentially expressed in the polypoid dysplasia of 8 mg FA-treated animals, as well as in colon lesions from animals treated with 0 mg FA. In fact, 12 out of the 15 genes in Table 1 were found to be differentially expressed when the mRNA expression profile of polypoid dysplasia vs. non-dysplastic colon mucosa from animals in the 0 mg and 8 mg groups was compared. These data suggest that the genes modified by FA supplementation in the non-dysplastic mucosa are also associated with colon tumorigenesis. As is evident from the heatmaps and immunohistochemical staining of β-catenin (Figure 10 and Figure 11), a dose-dependent increase in β-catenin signaling was observed in polypoid dysplasia following FA supplementation. Although not all the genes in Table 1 have been studied in colitis-associated CRC, the pathways in which they participate, primarily Wnt/β-catenin signaling, cell proliferation, and differentiation, represent known critical events in CRC formation. Enhanced expression of Dusp4 and Epha2, members of MAPK signaling, following FA supplementation, is consistent with the results of case-control studies which demonstrate that FA can interact with multiple members of the MAPK pathway [61]. These data suggest that the impact of FA supplementation on DNA methylation and gene expression occurs early and persists throughout the development of polypoid dysplasia.

In addition to modulating DNA methylation, FA supplementation increased the expression of inflammatory mediators. Results from a FA supplementation trial (1 mg FA daily for 8 weeks) of 10 high-risk subjects, with either a personal history of sporadic colorectal adenoma or a first-degree relative with sporadic CRC, suggested that high FA intake may promote the progression of CRC by enhancing inflammation and immune responses [62]. The concentration of folate in serum and colon tissues increased 50–80% over the course of the study, as compared to the baseline. Corresponding elevations in the expression of 642 genes associated with immune and inflammatory processes/responses (e.g., *IL6*, *IL17F*, *TNF*) were also observed following 4 and 8 weeks of FA treatment. Together, these data suggest that FA supplementation may enhance tumorigenesis by upregulating proinflammatory cytokines. 

## 5. Conclusions

The results from the present study indicate that FA promotes colitis-associated CRC by hypomethylating genes involved in key oncogenic pathways. Additional studies, including clinical trials, are warranted to further evaluate the potential for FA supplementation to increase CRC risk when administered to cancer-free patients with ulcerative colitis.

## Figures and Tables

**Figure 1 cancers-15-02949-f001:**
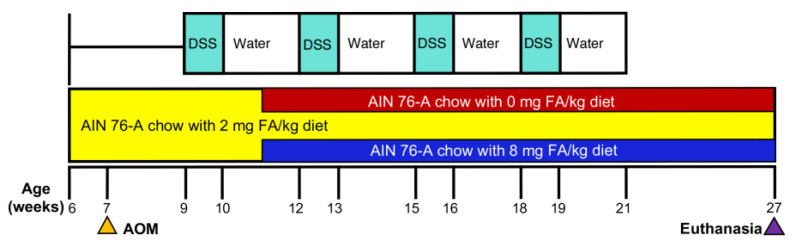
Experimental design of the study. Mice were injected with azoxymethane (AOM, 7.4 mg/kg i.p.) at 7 weeks of age. Colitis was induced with DSS, starting at 9 weeks of age. Mice were randomly assigned to receive different amounts of folic acid (FA) in an AIN-76A diet at 11 weeks of age and euthanized at 27 weeks of age (17–19/group).

**Figure 2 cancers-15-02949-f002:**
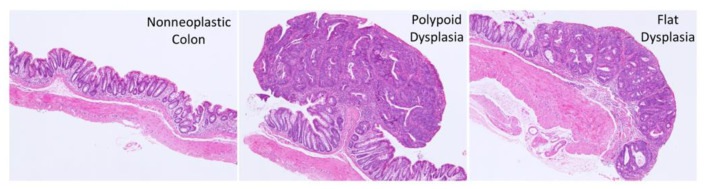
Low power view of non-neoplastic colon and dysplasias in the AOM/DSS model.

**Figure 3 cancers-15-02949-f003:**
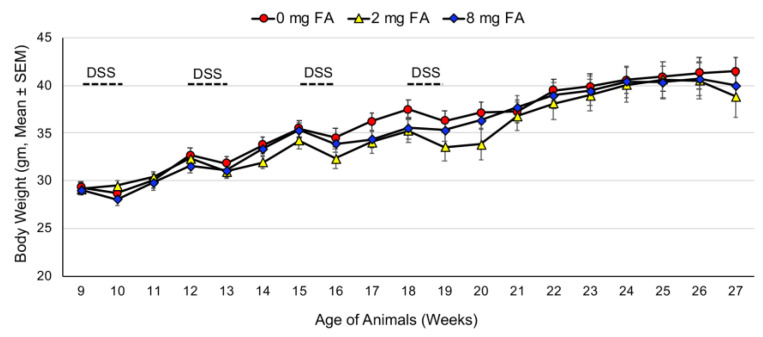
Weekly body weights of Swiss Webster mice with AOM/DSS-induced colitis. Mice were maintained on an AIN-76A diet supplemented with an either 0, 2, or 8 mg FA/kg diet (17–19/group).

**Figure 4 cancers-15-02949-f004:**
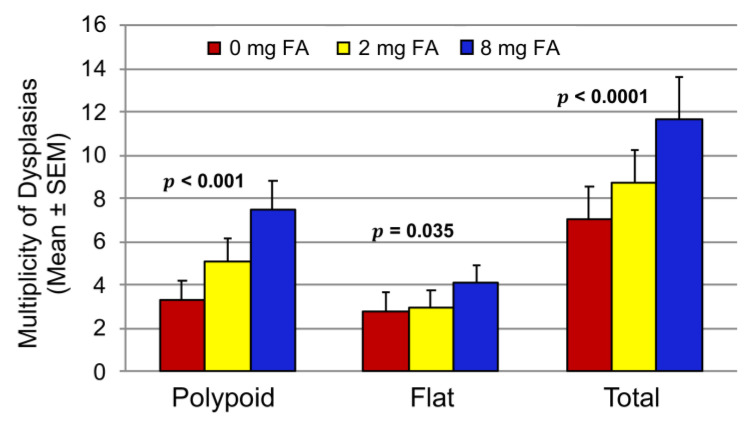
Effects of various doses of FA on the multiplicity of colitis-associated dysplasia. The total number of dysplasias is the sum of polypoid and flat dysplasias. Poisson regression with log link was used to determine the association between FA dose and tumor multiplicity (*n* = 17–19/group).

**Figure 5 cancers-15-02949-f005:**
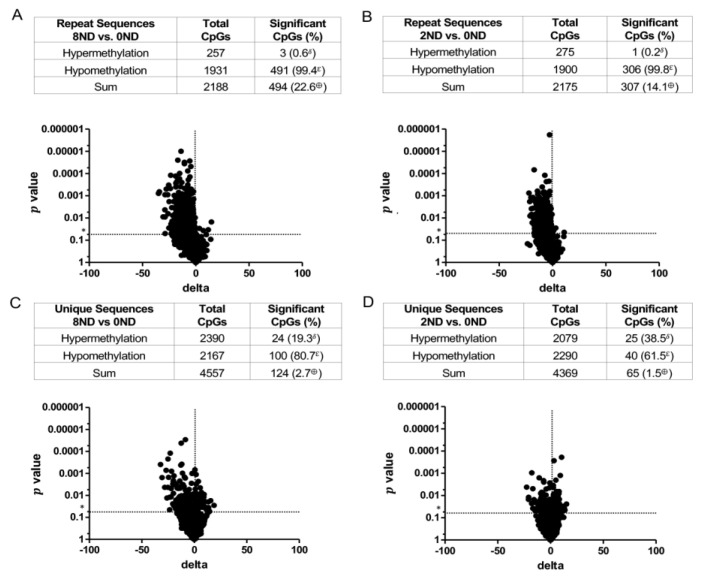
Volcano plots of DNA methylation profiles for LMD **non-dysplastic colonic epithelium** from **AOM/DSS-treated mice** receiving varying amounts of FA. **Repeat** sequences were compared in the (**A**) 8 mg FA- vs. 0 mg FA-treated epithelium (8ND vs. 0ND) and (**B**) 2 mg FA- vs. 0 mg FA-treated epithelium (2ND vs. 0ND). **Unique** sequences were compared in the (**C**) 8 mg FA- vs. 0 mg FA-treated epithelium (8ND vs. 0ND) and (**D**) 2 mg FA- vs. 0 mg FA-treated epithelium (2ND vs. 0ND). The *t*-test was used to compare site-specific DNA methylation levels between FA treatment groups (*n* = 8 per group). Sites with *p* values less than 0.05 (*) and more than a 5% methylation difference were considered significantly different between groups. δ = significant hypermethylated CpGs expressed as a percentage of the total sum of significant CpGs; ε = significant hypomethylated CpGs expressed as a percentage of the total sum of significant CpGs; ⊕ = sum of significant CpGs (hypomethylated and hypermethylated) expressed as a percentage of total CpGs.

**Figure 6 cancers-15-02949-f006:**
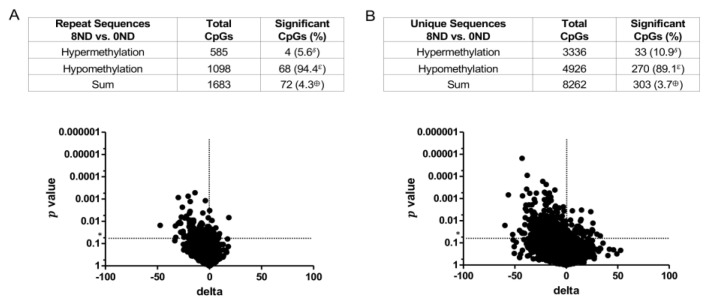
Volcano plots of DNA methylation profiles for LMD **non-dysplastic colonic epithelium plus the lamina propria** from **AOM/DSS-treated mice** receiving varying amounts of FA. (**A**) **Repeat** and (**B**) **unique** sequences of non-dysplastic colonic epithelium plus the lamina propria from mice in the 8 mg vs. 0 mg FA group (8ND vs. 0ND) were compared. The *t*-test was used to identify significant alterations in site-specific DNA methylation levels between FA treatment groups (*n* = 12 per group). Sites with *p* values less than 0.05 (*) and more than a 5% methylation difference were considered significantly different. δ = significant hypermethylated CpGs expressed as a percentage of the total sum of significant CpGs; ε = significant hypomethylated CpGs expressed as a percentage of the total sum of significant CpGs; ⊕ = sum of significant CpGs (hypomethylated and hypermethylated) expressed as a percentage of total CpGs.

**Figure 7 cancers-15-02949-f007:**
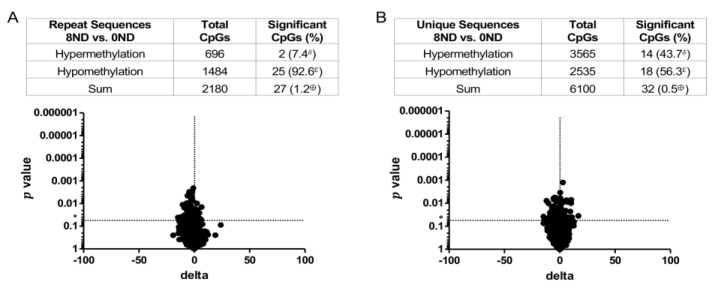
Volcano plots of DNA methylation profiles for LMD **non-dysplastic colonic epithelium from AOM-treated** (non-colitis) mice receiving varying amounts of FA. (**A**) **Repeat** and (**B**) **unique** sequences of non-dysplastic and non-inflamed colonic epithelium from the 8 mg vs. 0 mg FA group (8ND vs. 0ND) were compared. The *t*-test was used to identify significant alterations in site-specific DNA methylation levels between FA treatment groups (*n* = 7–8 per group). Sites with *p* values less than 0.05 (*) and more than a 5% methylation difference were considered significantly different. δ = significant hypermethylated CpGs expressed as a percentage of the total sum of significant CpGs; ε = significant hypomethylated CpGs expressed as a percentage of the total sum of significant CpGs; ⊕ = sum of significant CpGs (hypomethylated and hypermethylated) expressed as a percentage of total CpGs.

**Figure 8 cancers-15-02949-f008:**
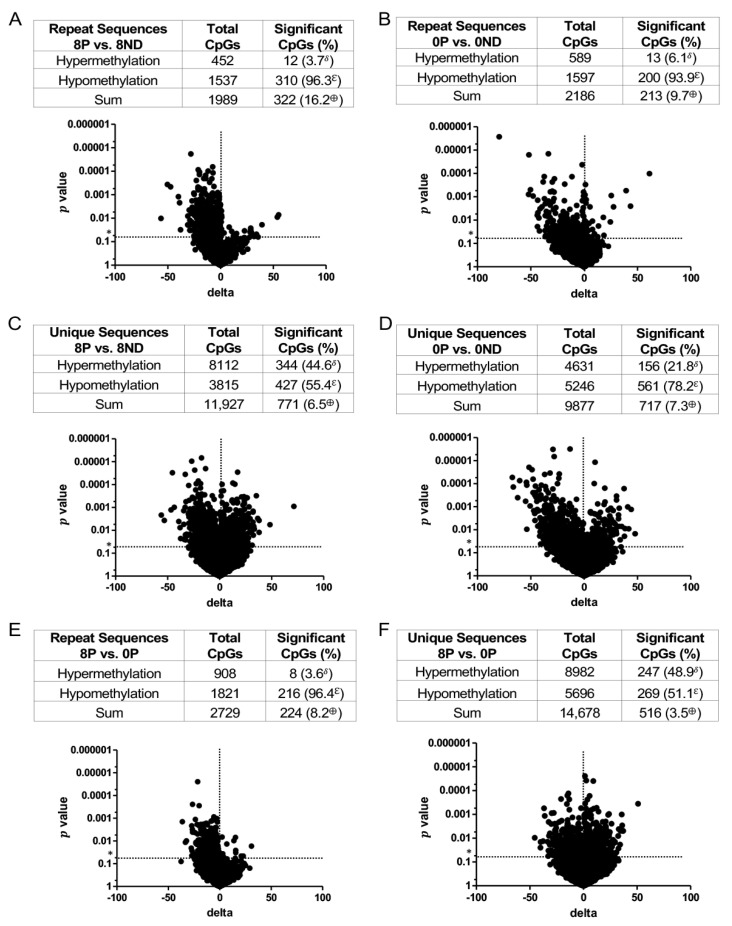
Volcano plots of DNA methylation profiles for **polypoid dysplasia** from **AOM/DSS**-treated mice receiving 0 or 8 mg of FA. (**A**) **Repeat** sequences: 8 mg FA-treated polypoid dysplasia vs. non-dysplastic epithelium (8P vs. 8ND); (**B**) **repeat** sequences: 0 mg FA-treated polypoid dysplasia vs. non-dysplastic epithelium (0P vs. 0ND); (**C**) **unique** sequences: 8 mg FA-treated polypoid dysplasia vs. non-dysplastic epithelium (8P vs. 8ND); (**D**) **unique** sequences: 0 mg FA-treated polypoid dysplasia vs. non-dysplastic epithelium (0P vs. 0ND); (**E**) **repeat** sequences: 8 mg FA vs. 0 mg FA-treated polypoid dysplasia (8P vs. 0P); and (**F**) **unique** sequences: 8 mg FA vs. 0 mg FA-treated polypoid dysplasia (8P vs. 0P). The *t*-test was used to compare site-specific DNA methylation levels between FA treatment groups (*n* = 12 per group for non-dysplastic epithelium and *n* = 8 per group for polypoid dysplasia). Sites with *p* values less than 0.05 (*) and more than a 5% methylation difference were considered significantly different between groups. δ = significant hypermethylated CpGs expressed as a percentage of the total sum of significant CpGs; ε = significant hypomethylated CpGs expressed as a percentage of the total sum of significant CpGs; ⊕ = sum of significant CpGs (hypomethylated and hypermethylated) expressed as a percentage of total CpGs.

**Figure 9 cancers-15-02949-f009:**
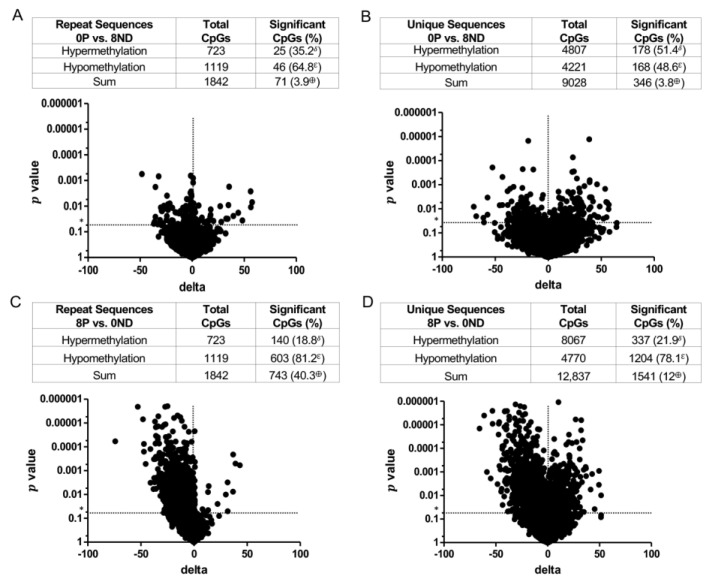
DNA methylation status is impacted by an interaction **between FA** dose and colon tissue type. Volcano plots summarizing the methylation profiles of (**A**) repeat and (**B**) unique sequences in 0 mg FA-treated polypoid dysplasia vs. 8 mg FA-treated non-dysplastic epithelium (0P vs. 8ND), and (**C**) repeat and (**D**) unique sequences in 8 mg FA-treated polypoid dysplasia vs. 0 mg FA-treated non-dysplastic epithelium (8P vs. 0ND). The *t*-test was used to compare site-specific DNA methylation levels between FA treatment groups (*n* = 12 per group for non-dysplastic epithelium and *n* = 8 per group for polypoid dysplasia). Sites with *p* values less than 0.05 (*) and more than a 5% methylation difference were considered significantly different between groups. δ = significant hypermethylated CpGs expressed as a percentage of the total sum of significant CpGs; ε = significant hypomethylated CpGs expressed as a percentage of the total sum of significant CpGs; ⊕ = sum of significant CpGs (hypomethylated and hypermethylated) expressed as a percentage of total CpGs.

**Figure 10 cancers-15-02949-f010:**
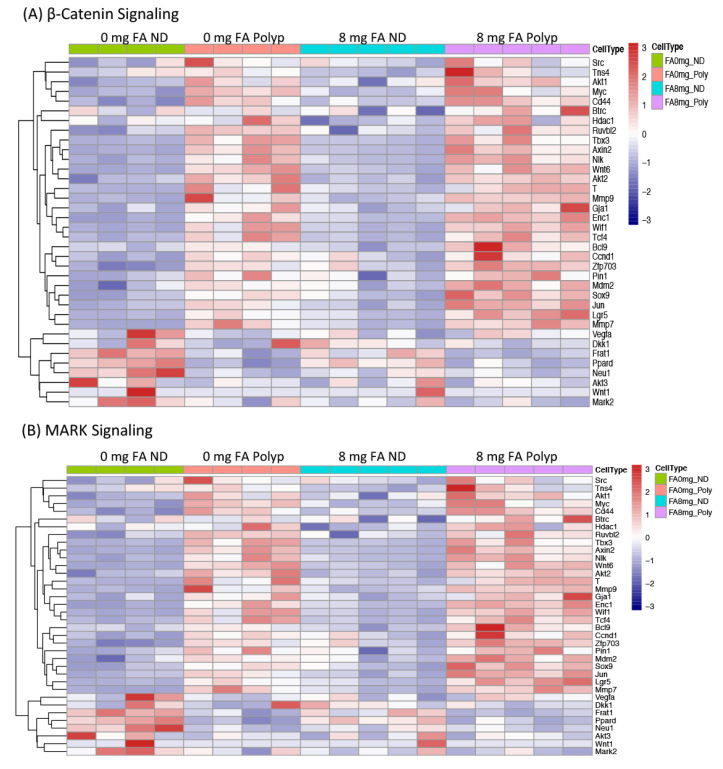
Heatmaps of the expression level of genes in the β-catenin and MAPK pathways. Mice with AOM/DSS-induced colitis were administered 0 mg or 8 mg FA. Epithelial cells from the non-neoplastic colonic mucosa or polypoid lesions were LMD and analyzed by RNA-Seq. The expression of genes associated with (**A**) β-catenin and (**B**) MAPK signaling were profiled for several treatment groups: 0 mg FA-treated non-dysplastic epithelium (0 mg FA ND), 0 mg FA-treated polypoid (0 mg FA Polyp), 8 mg FA-treated non-dysplastic epithelium (8 mg FA ND), and 8 mg FA-treated polypoid (8 mg FA Polyp).

**Figure 11 cancers-15-02949-f011:**
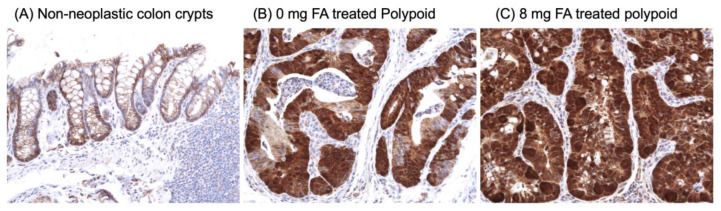
Immunohistochemical staining of β-catenin in the murine colon. Representative images (200×) show (**A**) membrane staining of β-catenin in the non-dysplastic colon and (**B**,**C**) overexpression and nuclear localization of β-catenin in polypoid lesions. Unlike polypoid lesions from mice treated with FA 0 mg, the majority of tumor cells in polypoid lesions from mice treated with FA 8 mg exhibited overexpression or nuclear localization of β-catenin.

**Figure 12 cancers-15-02949-f012:**
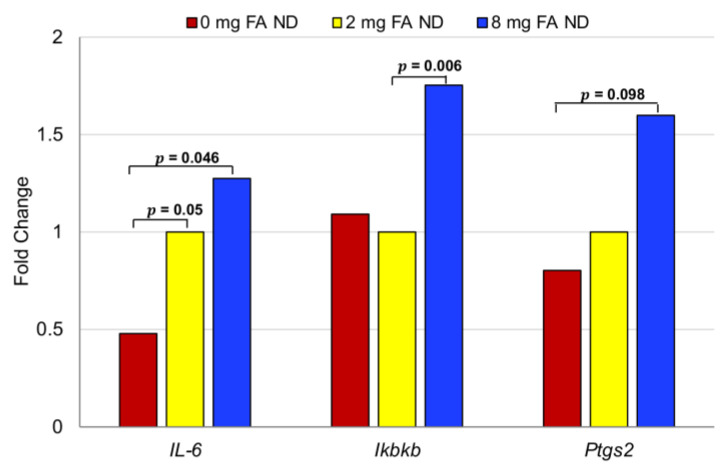
Effects of FA on the expression of inflammatory mediators. The expression level of inflammatory genes was evaluated using LMD non-dysplastic colonic mucosa (ND) from AOM/DSS-treated mice fed 0, 2, or 8 mg FA. Results are expressed as the fold-change in relative levels of each gene transcript for mice receiving varying levels of FA. Statistical analyses were performed using the *t*-test. *n* = 8 per group.

**Figure 13 cancers-15-02949-f013:**
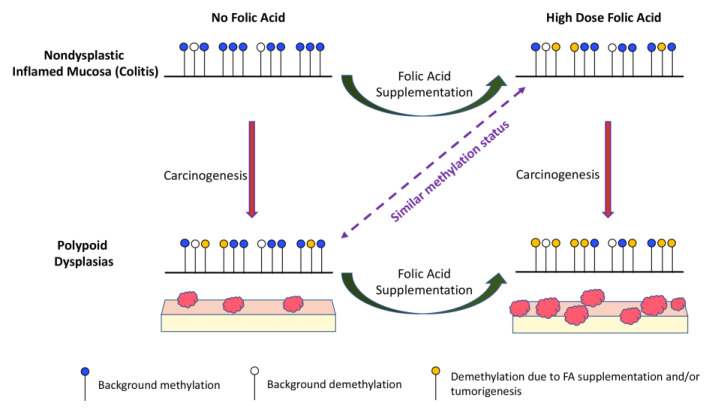
Proposed mechanism by which FA alters DNA methylation and promotes colitis-associated tumorigenesis.

**Table 1 cancers-15-02949-t001:** Genes Both Differentially Expressed * and Differentially Methylated Following FA Supplementation (8 mg/kg vs. 0 mg/kg diet).

A	B	C	D	E	F
Gene	ND ** FA 8 mg/0 mg	Polyp ^ζ^ FA 8 mg/0 mg	8 mg FA Polyp/ND	0 mg FA Polyp/ND	Signaling pathways and/or cellular function
Tbxt (T, Brachyury)	**11.8**	0.9	**4.7**	**61.4**	Wnt target gene [32,33]
Slc4a11	**3.1**	1.0	**3.2**	**10.5**	H^+^(OH^×^) and NH_3_-H^+^ transporter protein; [34] necessary for cell survival [35]
Dusp4 (Mkp-2)	**2.1**	1.4	**4.7**	**7.0**	Negative regulator of MAPK, overexpressed in CRC [36,37]
Spink4	**2.1**	1.1	**0.5**	0.9	Marker for intestinal goblet cells, downregulated in CRC [38,39]
Thbs1 (Tsp1)	**1.7**	1.1	**2.9**	**4.6**	Negative regulator of angiogenesis; [40] increased expression in inflammatory bowel disease [41]
Epha2	**1.8**	0.9	**2.1**	**4.1**	Effector of MAPK signaling and overexpressed in CRC [42,43]
Zfp703	**1.7**	1.3	1.5	**2.0**	Wnt target gene [44]
Enc1	1.4	1.2	**2.1**	**2.4**	Wnt target gene [45]
Atp11a	1.3	1.3	**2.8**	**2.6**	ATPase with higher expression in CRC [46]
Lzts1 (Fez1)	0.8	0.8	**2.0**	**2.1**	Tumor suppressor gene; regulates M phase during cell cycle [47,48]
Tns4 (Cten)	**0.6**	1.2	**2.6**	1.2	Interacts with β-catenin; overexpressed in CRC [49]
Neu1	**0.6**	1.1	0.8	**0.4**	Inhibits β-catenin expression [50]
Rab11fip4	**0.6**	0.9	**0.5**	**0.3**	HIF-1α target gene and effector of RAB11 [39]
Rims4	**0.5**	**0.3**	**0.2**	**0.1**	Maintains Ca^2+^ influx [51]
Fut9	**0.3**	1.1	0.8	**0.3**	Essential in tumor-initiating cells; switched off to enhance the aggressiveness of tumor cells [52]

* Values represent fold-change, as determined using both the Cuffdiff and DESeq2 methods. All values in **bold** are statistically significant (*p* ≤ 0.05). The remainder did not reach statistical significance using either method. *n* = 4–5 per tissue type per treatment group. ** ND = non-dysplastic mucosa; **^ζ^** Polyp = polypoid dysplasia.

## Data Availability

Raw data from the DNA methylation and RNA-Seq analyses have been submitted to the GEO repository. The GEO accession numbers are GSE168975 (DNA methylation) and GSE168525 (RNA-Seq).

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
