# Peer review of "Folic Acid Supplementation Promotes Hypomethylation in Both the Inflamed Colonic Mucosa and Colitis-Associated Dysplasia"

_cancers, 2023, doi:10.3390/cancers15112949_

Round 1
Reviewer 1 Report (Previous Reviewer 2)
Author addressed all the comments adequately.
Author Response
No comments need to be addressed.
Reviewer 2 Report (Previous Reviewer 3)
I have some comments about this re-submitted paper.
1. Line 176: please correct MilliporeSigma
2. Line 206: since no significant difference was found (Fig3), please correct/remove the sentence: "mice fed with 2 mg FA had the lowest body weight..."
3. Line 383: reference for Fig10 is missing from the main text.
4. Figure 11: this is a critical figure in the manuscript. Thus, please provide quantitative data (e.g. count the ratio of colon/tumor cells with nuclear beta-catenin), not only representative images.
5. Figure 12, legend, line 429: "Expressions of Statistical analysis" -please correct this statement.
Author Response
- Line 176: please correct MilliporeSigma
Response: This error has been corrected. It has been changed to Sigma-Aldrich (line 176).
- Line 206: since no significant difference was found (Fig 3), please correct/remove the sentence: “mice fed with 2 mg FA had the lowest body weight…”
Response: We thank the reviewer for this suggestion. The sentence has been modified with removal of “mice fed with 2 mg FA had the lowest body weight…” (lines 205-206).
- Line 383: reference for Fig 10 is missing from the main text.
Response: We thank the reviewer for this comment. Citation of Fig 10 has been added into the main text (line 390).
- Figure 11: this is a critical figure in the manuscript. Thus, please provide quantitative data (e.g., count the ratio of colon/tumor cells with nuclear beta-catenin), not only representative images.
Response: We thank the reviewer for this suggestion. Sentence has been revised to include the percentage of cells with nuclear beta-catenin (lines 396-398). Legend of Figure 11 was revised as well.
- Figure 12, legend, line 429: “Expressions of Statistical analyses” – please correct this statement.
Response: This error has been corrected. “Expressions of” has been deleted (line 442).
This manuscript is a resubmission of an earlier submission. The following is a list of the peer review reports and author responses from that submission.
Round 1
Reviewer 1 Report
In this report, the authors aim to investigate the effect of folic acid (FA) supplementation on colitis-associated colorectal cancer (CAC) using the azoxymethane/dextran sulfate sodium (AOM/DSS)-model of induced CAC .
The present study addresses an important problem in ulcerative colitis as a major risk factor for developing colorectal cancer (CRC) in regards to the use of supplementary folic acid. Results from several studies indicate that the risk of developing CRC is decreased in patients taking FA supplements. In contrast, some other works showing that FA supplementation does not protect against CRC.
The methodological approach is adequate. The authors included a reasonable number of experiments including time and dose response results as well as controls. Histopathological analysis, isolation of DNA and RNA from micro dissected cells, alteration in DNA methylation, comparison of gene expression changes, pathway analyses of genes were well performed, well described and results were well shown.
Overall, this is an excellent, clear, well written work and deserves publication in Cancers.
Reviewer 2 Report
To assess the effect of folic acid (FA) supplementation on colitis-associated colorectal cancer (CAC) using the azoxymethane/dextran sulfate sodium (AOM/DSS)-model of induced CAC.
In general, author made a very good attempt to prove the hypothesis. However, there are few questions needs to be clarified.
Major comments:
1. Fig 3, It would be important to show the IHC staining images with the positive and negative controls.
2. Based on the results, it is clear that FA promotes colitis-associated CRC by hypomethylating genes involved in key oncogenic pathways. In addition, author showed the alteration of gene expression by FA. However, author did not show the molecular mechanism how this FA impact the CAC.
3. Also, author indicated the effects of FA on methylation status of genes involved in Wnt/β-catenin and MAPK signaling, did it have effects on protein levels? Please clarify.
Minor Comments:
1. Figure 2: It would be good and important to indicate the units for the body weight.
2. In addition to the table 1, It would be good to show as a pathway of differentially expressed and differentially Methylated genes following FA, this would add more scientific weightage and also very easy to understand by the audience.
Reviewer 3 Report
Folic acid (FA) supplementation is frequently recommended for colitis patients, however, it may be a risk factor for colorectal cancer (CRC) development. Furthermore, mouse experiments with FA led to contradictory results and suggested that FA may have a different effect when applied before or after the onset of tumorigenesis. In thsi manuscript, Chang WCL et al study changes in the methylation of normal colon and dysplastic epithelium in a colitis-associated, DSS-induced colorectal cancer (CRC) mouse model in the presence/absence of different doses of FA. They applied laser-dissected tissue for methylation analysis, RNA-Seq and RT-qPCR to prove the role of FA and to identify the mechanism behind the effect of this compound. Thus, this work addresses an important scientific question.
My major comments:
1. The authors study inflammatory markers in their model. However, results from these experiments are inconclusive and they only found a change for IL-6. The authors should carry out a wider screen for inflammatory genes not only in normal, but also in dysplastic tissues. How do the expression levels of these genes change in dysplasia? How do other genes involved in inflammation behave?
2. Since there is often no one-to-one correlation between RNA and protein data, the authors should prove their inflammatory marker results at the protein level, too (e.g. immunohistochemistry).
Minor comments:
3. Fig. 4-8: please indicate the percentage values in the figure panels.
4. Lines 287-316: please indicate the exact percentage values in the figures, and not in the text. Please shorten the text, this is very difficult to read in its present form full with percentage values.
5. Chapter 3.5: please show all RT-qPCR data in a figure, not in a table.
6. Table 1: please show at least the most important data on a graph, too.
7. Please reduce the discussion (e.g. lines 471-475 are simple repetitions from the results section).